# Wild vs. Cultivated *Zingiber striolatum* Diels: Nutritional and Biological Activity Differences

**DOI:** 10.3390/plants12112180

**Published:** 2023-05-31

**Authors:** Jing Yang, Yaochen Li, Yuxin He, Hongying He, Xiaoqi Chen, Tingfu Liu, Biao Zhu

**Affiliations:** 1Key Laboratory of Quality and Safety Control for Subtropical Fruit and Vegetable, Ministry of Agriculture and Rural Affairs, Collaborative Innovation Center for Efficient and Green Production of Agriculture in Mountainous Areas of Zhejiang Province, College of Horticulture Science, Zhejiang A&F University, Hangzhou 311300, China; yangjing@zafu.edu.cn (J.Y.); liyaochen@stu.zafu.edu.cn (Y.L.); hhy_0801@163.com (H.H.); chenxq@zucc.edu.cn (X.C.); 2School of Information and Electrical Engineering, Hangzhou City University, Hangzhou 310015, China; 3Lishui Academy of Agricultural Sciences, Lishui 323000, China

**Keywords:** *Zingiber striolatum*, nutritional value, chemical compounds, biological activity

## Abstract

Compositional, functional, and nutritional properties are important for the use-value assessments of wild and cultivated edible plants. The aim of this study was to compare the nutritional composition, bioactive compounds, volatile compounds, and potential biological activities of cultivated and wild *Zingiber striolatum*. Various substances, such as soluble sugars, mineral elements, vitamins, total phenolics, total flavonoids, and volatiles, were measured and analyzed using UV spectrophotometry, ICP-OES, HPLC, and GC-MS methods. The antioxidant capacity of a methanol extract of *Z*. *striolatum*, as well as the hypoglycemic abilities of its ethanol and water extracts, were tested. The results showed that the contents of soluble sugar, soluble protein, and total saponin in the cultivated samples were higher, while the wild samples contained higher amounts of K, Na, Se, vitamin C, and total amino acids. The cultivated *Z*. *striolatum* also showed a higher antioxidant potential, while the wild *Z*. *striolatum* exhibited a better hypoglycemic activity. Thirty-three volatile compounds were identified using GC-MS in two plants, with esters and hydrocarbons being the main volatile compounds. This study demonstrated that both cultivated and wild *Z. striolatum* have a good nutritional value and biological activity, and can be used as a source of nutritional supplementation or even in medication.

## 1. Introduction

Wild plant resources play an indispensable role in the history of human development, especially in maintaining people’s balanced diet and the medicine in resource-poor areas. Some studies have highlighted the important beneficial effects of wild species on health. These effects are mainly related to the high contents of some secondary metabolites with biological activity, such as flavonoids [1], phenolic acids [2], and saponins [3]. In recent years, people have become more and more aware of the health benefits of natural products and the demand for functional foods is increasing, which makes wild edible vegetables the focus of attention.

*Zingiber striolatum* belongs to the *Zingiber* species and is a special medicinal and food plant resource in China [4]. It is widely distributed, mainly in its wild form, and less so in its cultivated form. Its edible part is its flower, which has an aromatic odor and can be made into dried fruit and steamed, stir-fried, or eaten cold. *Z. striolatum* has attracted much attention due to its multiple functional properties and medicinal value. At present, more than 100 *Z. striolatum*-related patents can be found in the China Intellectual Property Office, and it has a very large market prospect. Several phytochemical studies have shown that the ethanol extract of *Z. striolatum* has significant hypoglycemic activity in vitro [5]; the chloroform extract has a powerful inhibitory effect on free radical generation [6]; and Tian et al. [7] revealed that *Z. striolatum*’ volatile oil has strong antibacterial and anticancer activities. According to the “Compendium of Materia Medica”, *Z. striolatum* has a good medicinal value and has been used to promote blood circulation, dissolve phlegm, relieve cough, and reduce swelling and pain [4].

Recent studies have emphasized the commercial development potential of wild edible plants, and artificial cultivation has become inevitable to meet the enormous demand for wild vegetables in the market. Despite increasing reports on the biological activities and health benefits of wild edible plants, more research is still needed to reveal the effects of artificial cultivation on these plants before suggesting their commercial cultivation. Most existing reports have mainly focused on the wild edible species collected directly and determined their phytochemical and biological activities, without much comparative research on wild and cultivated species. Therefore, this report emphasizes the nutritional value, volatile species, safety, and application potential of the biological activity value of cultivated and wild *Z. striolatum*.

## 2. Results

### 2.1. Chemical Composition

#### 2.1.1. Moisture, Soluble Protein, and Soluble Sugars

Figure 1 shows the moisture (a), soluble protein (b), and soluble sugar (c) values of wild and cultivated *Z. striolatum*. The moisture content of the cultivated plants was higher than that of the wild plants, with no significant difference. In the wild *Z. striolatum*, the content of soluble protein ranged from 0.83 mg/g DW to 1.22 mg/g dry weight (DW), with an average value of 1.04 mg/g DW, which was significantly lower than the cultivated *Z. striolatum* (which ranged from 1.82 mg/g DW to 1.91 mg/g DW). Similarly, the soluble sugars, including glucose, fructose, and a small amount of sucrose, exhibited significantly higher values in the cultivated *Z. striolatum* compared to the wild plants. Furthermore, the main soluble sugars in both the wild and cultivated *Z. striolatum* were fructose and glucose.

#### 2.1.2. Minerals, Potentially Toxic Elements and Nitrate

The concentrations of the mineral elements in the wild and cultivated *Z. striolatum* are shown in Table 1. The order of the amount of each element differed slightly in different plants, namely K > Ca > P > Mg > Na > Mn > Fe > Zn > Cu > Se for the wild plants and K > P > Ca > Mg > Mn > Na > Fe > Zn > Cu > Se for the cultivated plants. There were significant differences in the concentrations of all ten mineral elements. On average, the concentrations of K (1.02-fold), Na (1.39-fold), and Se (3.43-fold) were higher in the wild material, while the others were higher in the cultivated material, including Ca (1.03-fold), Mg (1.18-fold), P (1.23-fold), Cu (1.23-fold), Fe (1.54-fold), Mn (1.86-fold), and Zn (1.33-fold). There was a close correlation between Na and K in the constant elements, and the sodium–potassium ratio (Na/K) is considered to be one of the most important indicators for the assessment of cardiovascular disease risk and cardiovascular-disease-related mortality. The World Health Organization (WHO) recommends that, ideally, humans should consume less than 2000 mg of sodium and more than 3510 mg of potassium per day and states that Na/K ≤ 1.0 is optimal for the protection of cardiovascular health [8]. Our analysis showed that the Na/K ((0.010) of the wild *Z. striolatum* was slightly higher than that of the cultivated *Z. striolatum* (0.007). In general, the Na/K of both materials were less than 1.0, indicating a better advantage in this regard.

There are different standards for the contaminant limits in vegetables about the potentially toxic elements in different regions. Various laws and regulations cover several important toxic metal elements, and it is crucial to verify them because of their hazardous nature. The results in this work showed that the Cr and Pb levels in the wild materials were below the corresponding thresholds for the highest levels of food contaminants (fresh vegetable, Cr ≤ 0.5 mg/kg FW and Pb ≤ 0.1 mg/kg FW) in China (GB 2762-2017); however, the Cr and Pb content of the cultivated *Z. striolatum* were 1.15-fold and 3.92-fold higher than the standard, respectively. The reason for these excess contaminants may be due to the greater ability of the species to retain this metal. In the case of As and Cd, when compared with the Chinese food contaminant limits for fresh vegetables (0.5 mg/kg FW and 0.05 mg/kg FW), both materials were within the normal range and their Cd contents were even lower than the Commission Regulation (EU) 2021/1317 contaminant limits (0.03 mg/kg FW).

The nitrate content in the wild material ranged from 7.16 mg/g DW to 8.13 mg/g DW, while in the cultivated material, it ranged from 6.68 mg/g DW to 7.35 mg/g DW; the nitrate content in the wild *Z. striolatum* was significantly higher. There is no specifical standard for flower bud vegetables yet. According to the GB19338-2003 limits for nitrate in vegetables, the nitrate content in both plants was higher than the standard for nitrate in stem vegetables (≤1200 mg/kg FW); compared with the (EU) 1258/2011 limits for nitrate in fresh lettuce (≤3000 mg/kg FW), the nitrate contents were lower than the limits. Previously, nitrate was considered to be a precursor of N-nitroso compounds, which are classified as human carcinogens. However, the International Agency for Research on Cancer (IARC) also stated that there is no substantial evidence to suggest that nitrate is an animal carcinogen [9]. Meanwhile, it has also been shown that the human body will convert nitrate to nitrite after ingestion, and part of the nitrite in human body might be converted to nitric oxide under low oxygen conditions, which could be harmful in excess [10]. Therefore, there may be some risk when consuming the two types of *Z. striolatum,* due to possible excessive nitrate intake.

#### 2.1.3. Free Amino Acid

Amino acids are essential for protein formation and have multiple functions in the human body, including essential and non-essential amino acids. *Z. striolatum* contains 7 essential amino acids (EAA) and 10 non-essential amino acids (NEAA) (Table 2 and Figure 2). Table 2 shows that the wild *Z. striolatum* (11.592 ± 0.383 mg/g DW, 12.85%) had higher levels of total amino acids (TAA) and EAA proportions compared to the cultivated *Z. striolatum*. 

Among the 17 amino acids detected, histidine (wild 34.28% and cultivated 20.70%), serine (wild 18.97% and cultivated 22.56%), aspartic acid (wild 9.23% and cultivated 13.28%), glutamic acid (wild 6.94% and cultivated 14.39%), and proline (wild 8.80% and cultivated 6.34%) had higher proportion levels in both of the *Z. striolatum* plants. Branched-chain amino acids (leucine and isoleucine) play a role in muscle synthesis [11] and are also highly accumulated in wild species. However, it is interesting to note that the contents of glutamic acid and arginine were higher in the cultivated *Z. striolatum*. In addition, the highest content in this experiment was histidine, which is inconsistent with the results of Zhang et al.’s [12] study on *Z. striolatum*. This may be due to the different sample origins and test methods.

#### 2.1.4. Vitamins

Vitamin C (VC) is an important biochemical compound found in many vegetables and fruits, with significant nutritional value and well-known antioxidant properties [13]. Meanwhile, vitamin E (VE) is the main component of lipophilic antioxidants in humans. Seeds and nuts are considered to be rich sources of VE compounds and some orange, red, and leafy green vegetables have also been found to contain abundant VE [14]. The contents of VC and α-tocopherol in the wild and cultivated *Z. striolatum* are shown in Figure 3a,b. The content of vitamin C in the wild varieties was significantly higher than that in the cultivated varieties. However, the VE contents did not show an obvious difference between the wild and cultivated plants.

#### 2.1.5. TP (Total Phenols) and TF (Total Flavonoids)

Plants are a source of natural bioactive compounds and secondary metabolites, most of which are of commercial importance, especially phenolic acids and flavonoids, which have a wide range of biological activities, including antioxidant, anti-inflammatory, antidepressant, and anticancer effects, which are important for plant and human health [15].

The cultivated *Z. striolatum* contained more total phenol content (12.3 ± 0.19 mg/g DW), but there was no significant difference between the two materials (Figure 4a). A similar situation was found in the total flavonoids content, as shown in Figure 4b.

#### 2.1.6. TPS (Total Polysaccharides) and TS (Total Saponins) 

Polysaccharides have been reported not only to be essential substances constituting life, but also to be one of the abundant natural products with various biological activities such as immunomodulatory, antitumor, antiviral, antioxidant, and anti-inflammatory activities [16]. Saponins are found in plants and some marine organisms, and have extensive biological and pharmacological properties, as well as being the main active ingredients in folk medicines, especially traditional Chinese medicine [17].

The cultivated materials were slightly higher than the wild ones in terms of their contents of TPS (Figure 5a) and TS (Figure 5b). The contents of TPS and TS in the wild plants were 11.17 ± 1.18 mg/g DW and 18.04 ± 0.04 mg/g DW, respectively, while they were 12.30 ± 0.19 mg/g DW and 21.97 ± 0.84 mg/g DW, respectively, in the cultivated plants. This means that *Z. striolatum* may also have a similar potential activity value to some Chinese herbal medicines with saponins [18]. At the same time, it is an anti-nutritional factor and is one of the endogenous agents that causes food poisoning [19]. The favorable and adverse effects of the saponins in *Z. striolatum* need to be further studied.

### 2.2. Bioactive Properties

#### 2.2.1. Antioxidant Activity

The results of the antioxidant activity in terms of the DPPH scavenging capacity, ABTS scavenging capacity assay, and iron-ion-reducing power assay (FRAP) are shown in Table 3. In the three systems analyzed, both the wild and cultivated plants showed a quite good antioxidant capacity. There was no significant difference between them except for FRAP, but the cultivated plants exhibited slightly higher values.

#### 2.2.2. α-Glucosidase and α-Amylase Inhibition

Clinically, α-glucosidase and α-amylase inhibitors have been shown to be effective agents that can inhibit postprandial glucose (especially in type 2 diabetes), but these inhibitors produce unwanted side effects and there is an urgent need to find new antidiabetic agents from natural sources to delay starch digestion [20]. Table 4 shows the results of the α-glucosidase and α-amylase inhibition with different solvent extracts of *Z. striolatum*. The ethanol extracts of the wild and cultivated *Z. striolatum* showed a higher inhibition of α-glucosidase (98.72% and 98.66%) than the inhibition of α-amylase (78.28% and 81.52%). There was no obvious differences between the cultivated and wild plants, but the hypoglycemic capacity of the ethanol extracts from both types of *Z. striolatum* was significantly higher than that of acarbose and 60% ethanol (mock). In contrast, the water extracts of the wild and cultivated *Z. striolatum* had a lower inhibitory capacity for α-glucosidase and α-amylase. The inhibition of α-glucosidase could not be seen, while the inhibitions of α-amylase were 53.26% and 43.97%, respectively, and all of them were even lower than acarbose. 

### 2.3. Volatie Composition

A comparative GC-MS analysis of the wild and cultivated *Z. striolatum* showed that they differed in their chemical compositions. A total of 33 volatiles were detected in the *Z. striolatum* (Table 5 and Appendix A), of which the wild *Z. striolatum* contained 23 species and the cultivated *Z. striolatum* contained 25 species. The main volatile substances detected in the wild *Z. striolatum* were palmitic acid ethyl ester, linoleic acid ethyl ester, linolenic acid ethyl ester, tricosane, and palmitic acid, etc.; the main volatile substances detected in the cultivated *Z. striolatum* were 1,1′-ethylenebisdecalin, linolenic acid ethyl ester, palmitic acid ethyl ester, β-pinene, and β-phellandrene, etc.

There were 15 volatile components shared by the wild and cultivated *Z. striolatum*, among which, α-pinene and β-pinene had a unique odor [21]. Among the 15 compounds, some were significantly more abundant in the wild *Z. striolatum*, such as palmitic acid (1.54-fold as much as cultivated plants) and palmitic acid ethyl ester (2.38-fold as much as cultivated plants). 

It is worth noting that eight compounds were present in the volatiles of the wild *Z. striolatum* that were not detected in the cultivated material, such as thujene, ethyl myristate, and linoleic acid ethyl ester, etc. Among the specific components (10 compounds) of the cultivated *Z. striolatum*, β-phellandrene, linoleic acid, and 1,1′-ethylenebisdecalin had higher levels, which may influence the odor of this variety.

From the perspective of the types of volatile compounds (Table 5 and Appendix A), the wild *Z. striolatum* contained five types of volatile compounds, namely esters, hydrocarbons, acids, alcohols, and ketones, among which, esters had the highest proportion (approximately 58.63% relative content), followed by hydrocarbons (approximately 16.27% relative content). The cultivated *Z. striolatum* also contained five types of volatile compounds, but no ketone compounds were found, with the highest proportion being ester compounds (approximately 24.60% relative content), followed by hydrocarbons (approximately 24.41% relative content). Other types of compounds accounted for a very high proportion, up to 19.66%. 

## 3. Discussion

*Z. striolatum* is an attractive edible wild plant because of its potential nutritional and medicinal property values. However, it remains an under-exploited and under-used plant. In this context, the focus of this study was to explore the variation in the nutritional, functional characteristics and volatiles of wild and cultivated types of *Z. striolatum*. Additionally, *Z. striolatum* was compared with some cultivated vegetables to reveal its potential preliminary value. In this study, the moisture of the cultivated *Z. striolatum* was higher than that of the wild *Z. striolatum*, which could be due to regular irrigation during its growth. Sugars have an important nutritional value for humans, being the main source of energy in plant foods and an important component of plant flavor components. The cultivated *Z. striolatum* also contained a higher soluble sugar content, and a higher sugar content in plants normally means a better taste; therefore, this may influence the acceptability of the plants to consumers, as has been reported in other studies [22,23]. The results obtained for soluble sugars were compared with those reported for lettuce [24] and soybean [25], and all three sugars were higher in both *Z. striolatum* than in lettuce, and the total sugars of the cultivars were even higher than some soybean varieties.

In addition to sugars, there were differences in the amino acids present in the samples. When plants are exposed to stress conditions, they accumulate a range of metabolites, especially amino acids. A report by Hayat et al. [26] showed a positive correlation between proline accumulation and plant stress. Additionally, the proline content of the wild *Z. striolatum* in our study was indeed much higher than that of the cultivar. Other explanations for this proline accumulation in cultivated species may be related to the developmental state of the plant at the time of collection [27]. We also found that branched-chain amino acids (BCAAs), such as leucine and valine, were more accumulated in the wild *Z. striolatum*. Bowne et al. [28] suggested that the increased levels of branched-chain amino acids observed in wheat were due to the drought experienced by the plants. Although we found that the wild *Z. striolatum* had less water content than the cultivated *Z. striolatum*, we cannot conclude that the wild *Z. striolatum* must have experienced drought, only that this was a possibility.

Other compounds that were observed in *Z. striolatum* include VC and VE. The results indicated lower levels of VC and VE in two types of *Z. striolatum* compared to some vegetables such as parsley and broccoli [29,30], but a higher VE content than that in fresh peas [31].

There was no significant difference in the total phenols and flavonoids between the two materials of *Z. striolatum*. Compared to common vegetables, the total phenol contents of both *Z. striolatum* were slightly lower than those in mustard leaves (19.7 mg/g DW) [31], but higher than those in kale (27.0 mg/100 g FW) [32], and their total flavonoid contents were also higher than those in asparagus (15.88 mg/g DW) [33]. Compared to *Zingiber officinale* [34], also a member of the ginger family, the *Z. striolatum* had lower total phenol and total flavonoid contents. This may be related to the different extraction procedures applied in the two studies, but also to the different solvents used for the hydroethanolic extracts [35]. However, the main reason is probably the differences in the plant genotypes. Usually, flavonoids are a type of phenolic compound and their contents are generally lower than those of total phenols, but different conclusions have been drawn in our study. We speculate that this phenomenon may be caused by different extractants. Mohammad et al. [36] studied the total phenols and flavonoids in Brassica vegetables using different solvents. The results showed that the total phenol content measured by the methanol extract of cauliflower was much lower than the total flavonoid content measured by the ethanol extract. Their results were similar to those from our study and may better explain this phenomenon.

Comparing the antioxidant activity of *Z. striolatum* with the study of Thilavech et al. [37], it was found that both the wild and culivated *Z. striolatum* had a higher DPPH scavenging capacity than cabbage and cauliflower (1.04 mg/g DW and 6.86 mg/g DW), but a significantly lower ferric-ion-reducing capacity, with values of 53.01 μmol/g DW and 209.95 μmol/g DW, respectively, for the cabbage and cauliflower. The reason for this difference may be that the compounds contained in methanolic extracts of wild greens may have poor metal ion chelating effects. Meanwhile, we found certain amounts of VC (wild 0.15 mg/g and cultivated 0.08 mg/g), total phenols (wild 11.07 mg/g and cultivated 12.30 mg/g), and total flavonoids (wild 16.26 mg/g and cultivated 17.31 mg/g) in the *Z. striolatum*, which have been shown to be associated with antioxidants in plants, as they are effective scavengers of free radicals, minimizing the harmful effects of oxidative stress [13,38]. Lockowandt et al. [39] also reported a significant correlation between the phenol acid content of *C. cyanus* plant parts (edible flowers and non-edible parts) and antioxidant activity. In this work, a higher DPPH scavenging activity, ABTS scavenging activity, and iron-ion-reducing power were found in cultivated *Z. striolatum*, and the total phenol and flavonoid contents of this material were also slightly higher than those of the wild plants. 

Most of the differences in the inhibitory effects of the two solvent-extracted *Z. striolatum* extracts were due to differences in the chemical functional compounds extracted from each solvent. However, the intake of acarbose can produce some side effects. In this case, medicinal plants may have a higher efficacy and less harmful effects than existing drugs. For this reason, the results we obtained revealed the potential value of *Z. striolatum*, which may be able to reduce the elevation of postprandial blood glucose levels in diabetic patients, as well as their ability to prevent type 2 diabetes [40]. Studies have shown that polysaccharides, polyphenols, kaempferol, quercetin, and saponins have a more or less similar potential to acarbose in inhibiting α-glucosidase and α-amylase [41], so we attribute the hypoglycemic activity of *Z. striolatum* to the phenol and flavonoid contents of the plant.

Wild vegetables are loved by people for their unique odor and flavor, and this special flavor is mostly due to the presence of special volatile substances. Among the volatiles shared by the wild and cultivated *Z. striolatum*, α-pinene has a citrusy, spicy, woody pine and turpentine-like aroma, which is reported to be high in lemon, Italian rosemary, and lavender, etc.; β-pinene is usually found together with α-pinene and tastes similar to α-pinene, but with a slight nuance of its menthol and camphor flavor [21]. Some studies have shown that the proportion of β-pinene is usually lower than that of α-pinene, but opposite result was found in the present study, which may have special research value for *Z. striolatum* [21]. For the specific composition of the cultivated *Z. striolatum*, α-phellandrene is the main component (>50%) of *Schinus molle* essential oil, which has been used in the food and perfume industries [42]; β-phellandrene has a citrus aroma [43]; and β-selinene has a pleasant, fresh, citrusy, menthol, and peppery odor (study of essential oil composition and antioxidant and antibacterial activity of duckweed), and also has antioxidant properties [44]. In addition to flavor, some substances in volatile matter have potential biological activity, which is also one of the reasons why people are more willing to accept wild vegetables. Some studies have shown that β-elemene can inhibit cell proliferation and induce apoptosis in a variety of cancers, including breast cancer and lung cancer [45]; α- and β-pinenes, which are present in many plant essential oils, and their pharmacological activities have been reported, including antibiotic resistance modulation and antitumor, antimalarial, antioxidant, anti-inflammatory, and analgesic effects [46]. Salas et al. [47] also found that both β-pinene and α- phellandrene have healing activity themselves, which can quickly heal wounds and make scars functional and aesthetically pleasing. Zhu et al. [48] reported that palmitic acid can inhibit the key molecules of PI3K/Akt, thereby preventing the proliferation and metastasis of prostate cancer.

## 4. Materials and Methods

### 4.1. Plant Material

The wild and cultivated *Zingiber striolatum* Diels were obtained from Li Shui, Zhejiang province (27°25′ N~28°57′ N, 118°41′ E~120°26′ E). Fresh buds were washed with ionized water, freeze-dried, crushed, and then stored at −20 °C until further analyses.

### 4.2. Chemical Composition Measurement

#### 4.2.1. Moisture

The samples were weighed (W1) before being freeze-dried for 72 h, and then they were weighed again (W2). Finally, the moisture percentage of a sample was calculated as:(1)$$Moisure\;(\%)=\frac{W1-W2}{W1}\times 100$$

#### 4.2.2. Soluble Protein

The soluble protein was determined using the Coomassie Brilliant Blue method. The extracts were prepared using 0.1000 g of freeze-dried powder and 8 mL of ultrapure water, using ultrasonic extraction for 15 min and centrifugation for 15 min at 8000 r/min. The supernatants were filtered and fixed to a 10 mL volumetric flask with ultrapure water. Then, 1 mL of the sample extract was mixed with 5 mL of Coomassie Brilliant Blue G-250 solution and left to stand for 5 min, before using a spectrophotometer (UV-2600, SHIMADZU, Tokyo, Japan) to detect the absorbance value at 595 nm. Then, the content of soluble protein was calculated according to the standard curve.

The standard solutions were prepared using a bovine serum protein standard solution at concentrations of 0 mg/mL, 0.012 mg/mL, 0.024 mg/mL, 0.048 mg/mL, 0.072 mg/mL, 0.084 mg/mL, and 0.1 mg/mL, respectively, and the absorbance value was detected at 595 nm, the same as the sample extracts. The standard curves were drawn with the absorbance value (A595) as the ordinate and the standard bovine serum protein concentration (mg/mL) as the abscissa.

#### 4.2.3. Soluble Sugar

The soluble sugars were determined using high-performance liquid chromatography coupled with a refraction index detector (HPLC–RID), mainly referring to the method of Yao [49]. The freeze-dried powder (0.1000 g) was mixed with 6 mL of ultrapure water in a water bath at 65 °C for 20 min. The extraction solution was filtered via a 0.45 μm microporous filter prior to the analysis.

An aliquot of extraction (5 μL) was analyzed in an Agilent 1200 HPLC system, using a Waters sugar-Pak I column WAT084038 (6.5 × 300 mm, 10 µm). The column temperature was maintained at 85 °C at a rate of 0.8 mL/min, and the mobile phase was ultrapure water. The contents of glucose, sucrose, and fructose in the samples were calculated according to the standard curves drawn from the results of the injection.

The mixed standard solutions of glucose, sucrose, and fructose were prepared at concentrations of 0 μg/mL, 50 μg/mL, 100 μg/mL, 200 μg/mL, 250 μg/mL, and 500 μg/mL, respectively, and detected the as same as the sample extracts. The standard curves were drawn with the peak area as the ordinate and the mixed standard solution concentration (mg/mL) as the abscissa.

#### 4.2.4. Minerals and Potentially Toxic Elements

The elements were determined as previously described by Yang et al. [50]. Briefly, the freeze-dried powder (0.5000 g) was digested with concentrated HNO_3_, fixed with water to 50 mL, and analyzed for minerals with an Inductively Coupled Plasma Optical Emission Spectrometer (ICP-OES). We used an IRIS/AP-ICP (TJA, USA) instrument and the ICP-OES procedure, mainly referencing the “Food National Standard for Determination of Multi-element in Food” (GB 5009. 268-2016).

#### 4.2.5. Nitrate

The nitrate content was determined by referring to Catado et al. [51]. The freeze-dried powder (0.1000 g) was extracted with 10 mL of ultrapure water in a water bath at 80 °C for 30 min and centrifuged at 8000 r/min for 10 min. The supernatants were filtered and fixed in a 10 mL volumetric flask with ultrapure water. A liquid of extraction (0.1 mL) was well mixed with 0.4 mL of 5% salicylic acid–sulfuric acid and left to stand for 30 min before 9.5 mL of 8% NaOH was slowly added. After the mixture was cooled to room temperature, the absorbance at 410 nm was determined and the content of the nitrate was calculated according to the standard curve.

The standard solutions of potassium nitrate were prepared at concentrations of 0.05 mg/mL, 0.1 mg/mL, 0.2 mg/mL, 0.25 mg/mL, and 0.5 mg/mL, respectively, and the absorbance value was detected at 410 nm, the as same as the sample extracts. The standard curves were drawn with the absorbance value (A410) as the ordinate and the concentration of potassium nitrate (mg/mL) as the abscissa.

#### 4.2.6. Free Amino Acid

The determination of the free amino acid was performed using the AccQ Tag system (Waters, Milford, MA, USA). The extracts were prepared using 0.1000 g of freeze-dried powder and 5 mL of ultrapure water in an ultrasonic bath at room temperature for 1 h. The extract was filtered through a 0.22 μm filter prior to separation using Waters Arc HPLC with an AccQ TagTM amino acid column (100 mm × 2.1 mm, 1.7 μm, Waters Corporation).

#### 4.2.7. Vitamin C

The vitamin C content was determined by referring to Bartoli et al. [52]. The freeze-dried powder (0.0500 g) was mixed with 5 mL of 20 g/L metaphosphoric acid in a 10 mL centrifuge tube and extracted using ultrasonication for 30 min. The supernatants were filtered through a 0.22 μm filter after being centrifuged at 8000 r/min for 10 min prior to the reduced ascorbic acid (AsA) analysis. 

A 10 μL aliquot of the sample was analyzed using an Agilent 1200 HPLC system (Agilent Technologies, Santa Clara, CA, USA) equipped with an InertSustain C18 column (4.6 mm × 250 mm, 5 µm) and a G1315B diode array detector (DAD) set at 254 and 265 nm. The mobile phase was 0.1% oxalic acid at a rate of 1 mL/min and the column temperature was 30 °C.

After the completion of the AsA measurement, dithiothreitol (DTT) was added to the injection bottle in a 1:1 volume ratio, and then reacted at 25 °C for 25 min prior to the total ascorbic acid (TAA) determination.

The contents of AsA and TAA were calculated according to the standard curve. The dehydrogenated ascorbic acid (DHA) content was obtained by subtracting the content of AsA from TAA.

#### 4.2.8. Vitamin E

The vitamin E content was determined by referring to Tang et al. [53]. The extracts were prepared using 0.2000 g of freeze-dried powder, 1 mL of a 1% BHT ethanol solution (2,6-di-tert-butyl-4-methylphenol), and 1 mL of a 100 g/L potassium hydroxide solution. The samples were extracted in an ultrasonic bath at 85 °C for 40 min. After cooling, 2 mL of petroleum ether was added in and shaken for 10 min before centrifuging at 8000 r/min for 10 min. The upper organic phase (ether layer) was transferred to a new test tube and washed three times with 3 mL of pure water. After that, 0.4 mL of the petroleum ether layer was evaporated in a fume hood and the residue was dissolved in 0.4 mL of methanol, filtered through a 0.22 μm filter, and then stored at 4 °C prior to analysis.

An aliquot of the extraction (20 μL) was analyzed in an Waters Arc HPLC system with an InertSustain C18 (4.6 mm × 250 mm, 5 µm). The rate was 1 mL/min and the mobile phases were methanol (mobile phase A) and ultrapure water (mobile phase B) (90:10). Additionally, the detection wavelength was 294 nm. The contents of α-tocopherol in the samples were calculated according to the standard curves drawn from the results of the injection.

The standard solutions of α-tocopherol were prepared at concentrations of 0 μg/mL, 10 μg/mL, 20 μg/mL, 40 μg/mL, and 50 μg/mL, respectively, and detected the as same as the sample extracts. The standard curves were drawn with the peak area as the ordinate and the concentration of α-tocopherol (μg/mL) as the abscissa.

#### 4.2.9. Total Phenol (TP), Total Flavonoid (TF), Total Polysaccharides (TPS), and Saponin (TS)

The TP content was determined using the Folin–Ciocalteau method, according to Colonna et al. [54], with some modifications. The extracts were prepared using 0.1000 g of freeze-dried powder and 10 mL of 80% methanol. Then, 0.6 mL of the extract, 3 mL of the Folin–Ciocalteu reagent, and 2 mL of 7.5% Na_2_CO_3_ were mixed and incubated for 60 min in the dark at room temperature. The total phenols content was determined with the absorbance at 765 nm with a spectrophotometer (UV-2600, SHIMADZU, Tokyo, Japan). The content of total phenols was expressed as equivalent mg of gallic acid per gram of the dry sample.

The determination of TF was performed following the methodology proposed by the AlCl_3_ colorimetric method [55]. Briefly, 20 mg/mL of the extract was prepared using 60% ethanol. The extract of 2 mL was mixed with 0.4 mL of AlCl_3_ (10%, p/v) and 0.4 mL of 5% NaNO_2_. The mixture was vortexed and incubated for 6 min before 4 mL of NaOH (4%) was added to stop the reaction. Finally, the absorbance at 510 nm was measured with a spectrophotometer (UV-2600, SHIMADZU, Tokyo, Japan), using 60% ethanol as a blank. The TF content was expressed as equivalent mg of rutin per gram of the dry sample.

The TPS content was determined by referring to Shang et al. [56]. The extracts were prepared using 0.2000 g of dried plant tissue and 4 mL of ethanol, and extracted using ultrasonication for 20 min. The supernatant was discarded after centrifuging and the insoluble pellet was washed twice with 4 mL of 80% ethanol. The residue was mixed with 4 mL of ultrapure water and extracted for 30 min at 40 °C in a water bath. The supernatant was taken for a polysaccharide content measurement after the extraction. Briefly, a 0.2 mL aliquot of the extract was well mixed with 1.6 mL of ultrapure water in a 10 mL centrifuge tube. Then, 1.0 mL of a 5% phenol solution and 5.0 mL of concentrated sulfuric acid were slowly added in order and incubated in a water bath at 30 °C for 20 min. Finally, the absorbance at 490 nm was measured with a spectrophotometer (UV-2600, SHIMADZU, Tokyo, Japan), using ultrapure water as a blank. The TPS content was expressed as equivalent mg of glucose per g of the dry sample.

The content of TS was quantified using the colorimetric method according to Le et al. [57], with some modifications. The extracts were prepared using 0.1000 g of dried plant tissue and 7 mL of 70% (*v*/*v*) ethanol, and sonicated at 55 °C for 40 min. In a test tube, 0.2 mL of the extract in a water bath at a temperature of 70 °C was evaporated, and 0.1 mL of 5% vanillin reagent and 0.4 mL of perchloric acid were added. Then, the reaction was heated in a water bath at 60 °C for 15 min, and ethyl acetate (4 mL) was added for 10 min. Finally, the absorbance at 560 nm was measured with a spectrophotometer (UV-2600, SHIMADZU, Tokyo, Japan), using 70% ethanol as a blank. The TS content was expressed as equivalent mg of oleanolic acid per gram of the dry sample.

### 4.3. Antioxidant Activity Evaluation

The antioxidant activity of the methanolic extracts was measured in vitro according to Colonna et al. [54].

The extract (10 mg/mL) was prepared using 80% ethanol in an ultrasonic bath at room temperature for 1 h. The supernatant was collected and the pellet was re-extracted twice, and all of the supernatants were combined and made up to a final volume of 25 mL.

DPPH radical scavenging activity: 2.7 mL of a DPPH methanol solution (0.1 mmol/L) was added to the methanol extract (0.3 mL) and the reaction was monitored at 517 nm until the absorbance was constant. The DPPH radical scavenging capacity was expressed as the equivalent amount of ascorbic acid (vitamin C) per gram of sample.

ABTS+ scavenging capacity: 3.8 mL of an ABTS+ working solution and extract (0.2 mL) were combined and the reaction monitored at 734 nm. The ABTS+ radical scavenging capacity was expressed as the amount of ascorbic acid (vitamin C) equivalent per gram of sample.

Iron-ion-reducing capacity: 1.8 mL of a TPTZ working solution (100 mL of 0.3 mol/L acetate buffer; 10 mL of 10 mmol/L TPTZ solution; and 10 mL of 20 mmol/LFeCl_3_ solution, a mixture of the three solutions) was added to 3.1 mL of ultrapure water and 0.1 mL of the extract to be tested, and the absorbance of the mixture was measured at 593 nm. The millimolarity of FeSO_4_ was expressed as its reducing power.

### 4.4. Hypoglycemic Activity Evaluation

The methanolic and water extracts for the hypoglycemic activity measurement in vitro were prepared as described by Chen et al. [55] and Shang et al. [56]. 

#### 4.4.1. Inhibitory Effect on α-Glucosidase Activity

The inhibitory effect of the extracts on the α-glucosidase activity was determined by referring to Lin et al. [58]. Briefly, 2.8 mL of PBS (0.2 M pH 6.8 phosphate buffer), 0.5 mL of a 0.1 U/mL α-glucosidase solution, and 0.5 mL of the extracts were mixed and incubated in a water bath at 37 °C for 10 min. Then, 0.5 mL of a 2.5 mM pNPG solution was added to the mixed solution and incubated for 40 min at 37 °C. Finally, 2 mL of a 0.2 M Na_2_CO_3_ solution was added to stop the reaction and the absorbance at 405 nm was measured.

The inhibition rate was calculated according to equation as follows, with acarbose and 60% ethanol being used as positive controls. Among them, A1 is the sample group, A2 is the sample control group, A3 is the blank group, and A4 is the blank control.
(2)$$\begin{array}{c} \mathrm{Inhibitory}\;\mathrm{activity}\left(\%\right)=\left(1-\frac{A1-A2}{A3-A4}\right)\times 100\% \end{array}$$

#### 4.4.2. Inhibitory Effect on α-Amylase Activity

The inhibitory effect of the extracts on the α-amylase activity was determined by referring to Zaharudin et al. [59]. Briefly, 0.5 mL of an α-amylase solution was mixed with 0.5 mL of the extracts and incubated in a water bath at 37 °C for 30 min. Then, 1 mL of a 2% starch solution was added. After another 15 min, 2 mL of a DNS reagent was used to terminate the reaction and the mixture was heated in a boiling water bath for 5 min and then cooled to room temperature. The absorbance at 540 nm was measured after adjusting the volume to 15 mL. The inhibition rate was calculated in the same way as in Section 4.4.1.

### 4.5. Volatie Composition Identification

The extracts were prepared using 0.1000 g of the freeze-dried powder and 2 mL of n-hexane, and extracted at room temperature for 4 h. After centrifuging at 10,000 r/min for 10 min, the extract was diluted and filtered through a 0.22 μm micro-porous membrane prior to the GC-MS analysis.

The GC operating conditions were as follows: a HP-5MS column (30 m × 0.25 mm × 0.25 µm); an injection temperature of 250 °C; and helium gas was used as the carrier gas at a flow rate of 1 mL/min. The initial column temperature was 50 °C and the temperature was ramped up to 250 °C at a rate of 10 °C/min, where it was held for 10 min.

The MS operating conditions were as follows: the interface temperature was 250 °C, the ion source temperature was 220 °C, the voltage was 70 eV, the detector voltage was 2106 V, and the mass range for the mass spectrometry scanning was 15~500 *m*/*z*.

According to the retention indices of the compounds, a comparative analysis was carried out using the NIST11.LIB database, and the relative content of each component was calculated using the area normalization method.

### 4.6. Statistical Analysis

The values of all the indexes determined in the experiment were recorded in triplicate and expressed as the mean ± standard deviation. SPSS Software (version 26.0, IBM SPSS Statistics), GraphPad Prism 9.0, and Enhanced data analysis software were used to analyze the data and draw the related graphs.

## 5. Conclusions

This study provides a detailed analysis and comparison of the nutritional composition, phytochemicals, antioxidant capacity, and in vitro hypoglycemic analyses of cultivated and wild *Z. striolatum* (Figure 6). The cultivated samples showed higher levels of soluble protein, soluble sugar, TPS, and TS, while the wild *Z. striolatum* showed higher levels of K, Na, and Se elements, and contained more amino acids and vitamin C. The two varieties had similar TP, TF, and antioxidant activities. However, in terms of their hypoglycemic ability, the wild *Z. striolatum* had a slight advantage.

In total, *Z. striolatum* contained a significant amount of essential dietary minerals and a variety of bioactive substances. In addition, we found that wild *Z. striolatum* (24 types) and cultivated *Z. striolatum* (26 types) contained a total of 33 volatile compounds, with compounds such as ethyl linolenate, α-pinene, and β-pinene sabinene having unique aromas. α-Pinene, sabinene, and palmitic acid also have various beneficial effects, such as antioxidant, anti-inflammatory, and anti-cancer activities. 

This study provides new data on the nutritional characteristics and bioactivity of *Z. striolatum*. Many people worry that wild varieties may be less safe than cultivated varieties, but from this trial, the wild varieties were not more dangerous in terms of potentially toxic elements. Meanwhile, based on our results, we suggest that the cultivation conditions of *Z. striolatum* need to be in an environment with relatively low heavy metal pollution. Despite some differences in their nutritional and biological activities, it can also be said that they are good sources of a wide range of nutrients and bioactive molecules. They have some potential in the processed food, nutritional products, and pharmaceutical industries, and further studies are needed subsequently to verify the feasibility of their application, while biocultural, environmental, and socioeconomic aspects should be considered if they are to be exploited sustainably.

## Figures and Tables

**Figure 1 plants-12-02180-f001:**
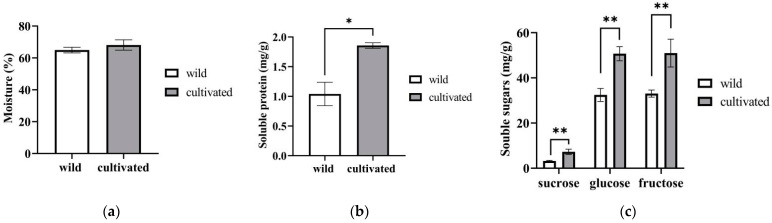
The contents of moisture (**a**), soluble protein (**b**), and soluble sugars (**c**) of wild and cultivated *Zingiber striolatum.* There is no significant difference in moisture of plants; “*” denotes significant difference at *p* < 0.05; and “**” denotes significant difference at *p* < 0.01.

**Figure 2 plants-12-02180-f002:**
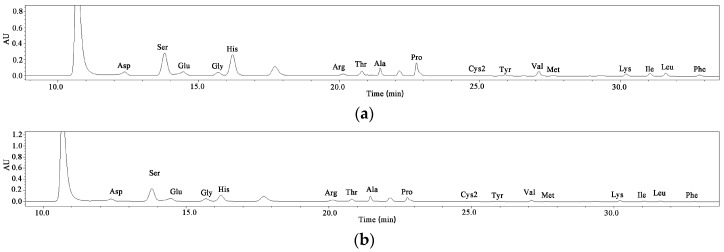
HPLC chromatograms of amino acids from wild (**a**), and cultivated (**b**) *Zingiber striolatum*.

**Figure 3 plants-12-02180-f003:**
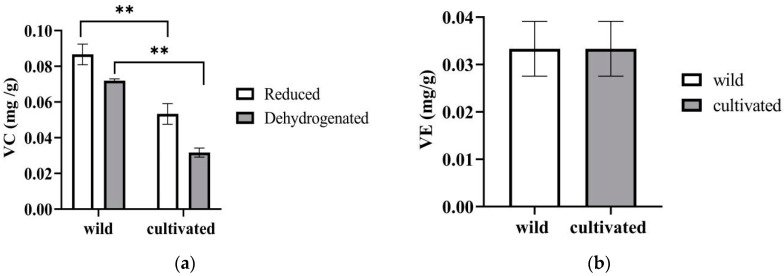
The contents of VC (**a**), and VE (**b**) from wild and cultivated *Zingiber striolatum.* “**” denotes significant difference at *p* < 0.01; there is no significant difference in VE of wild and cultivated *Zingiber striolatum*.

**Figure 4 plants-12-02180-f004:**
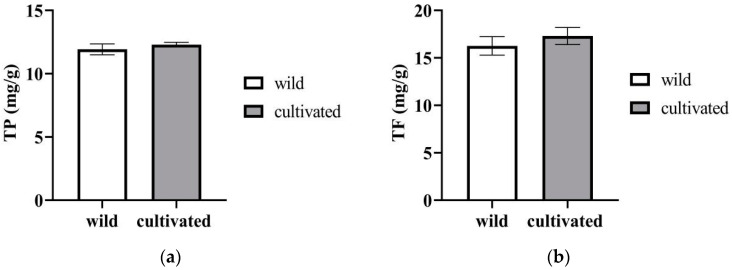
The contents of TP (**a**), and TF (**b**) of wild and cultivated *Zingiber striolatum.* There is no significant difference in TP and TF of wild and cultivated *Zingiber striolatum*.

**Figure 5 plants-12-02180-f005:**
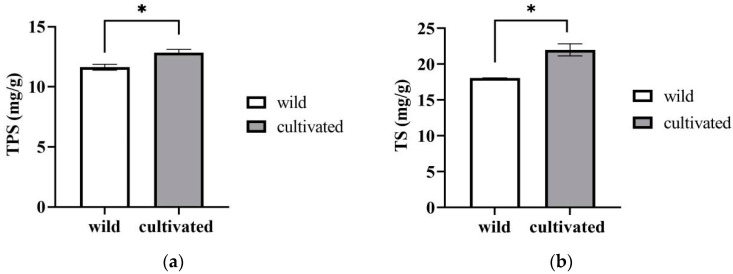
The contents of TPS (**a**), and TS (**b**) of wild and cultivated *Zingiber striolatum.* “*” denotes significant difference at *p* < 0.05.

**Figure 6 plants-12-02180-f006:**
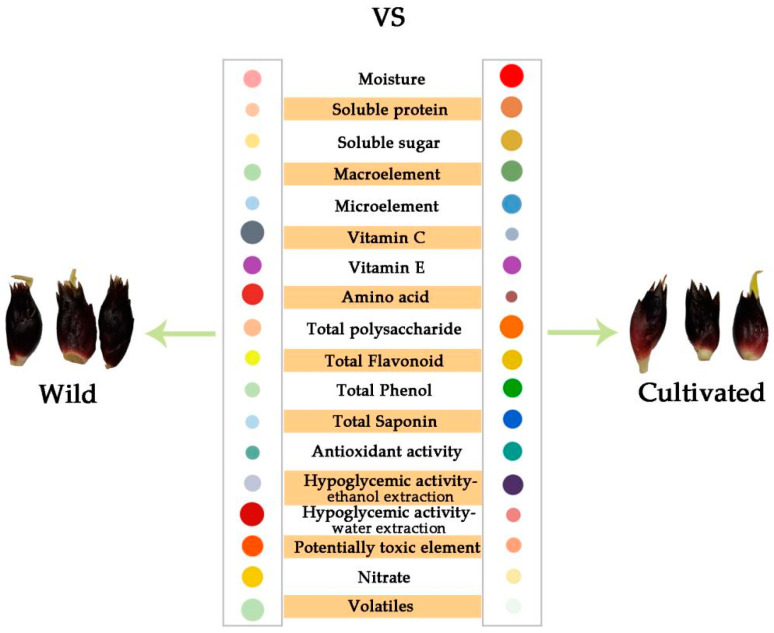
Relative comparison of compounds content and biological activity between wild and cultivated *Zingiber striolatum*. The sizes and shades of the circle represent the different relative levels of the parameter in wild and cultivated, namely the bigger size and deeper color indicate a higher level.

**Table 1 plants-12-02180-t001:** The contents of minerals, potentially toxic elements, and nitrate of wild and cultivated *Zingiber striolatum* (mg/100 g DW).

	Compounds	Wild	Cultivated
Minerals	Calcium (Ca)	596.38 ± 0.04 ^b^	613.20 ± 6.22 ^a^
	Potassium (K)	7091.76 ± 42.65 ^a^	6935.63 ± 41.63 ^b^
	Magnesium (Mg)	225.84 ± 1.52 ^b^	266.95 ± 1.15 ^a^
	Phosphorus (P)	508.15 ± 2.59 ^b^	622.77 ± 1.25 ^a^
	Sodium (Na)	67.74 ± 0.40 ^a^	48.60 ± 1.55 ^b^
	Cuprum (Cu)	0.70 ± 0.10 ^b^	0.86 ± 0.02 ^a^
	Ferrum (Fe)	7.57 ± 0.21 ^b^	11.64 ± 1.63 ^a^
	Manganese (Mn)	45.49 ± 0.14 ^b^	84.53 ± 0.62 ^a^
	Zinc (Zn)	7.35 ± 0.05 ^b^	9.75 ± 0.05 ^a^
	Selenium (Se)	0.07 ± 0.02 ^a^	0.02 ± 0.02 ^b^
Potentially toxic elements	Arsenic (As)	0.017 ± 0.007 ^a^	0.021 ± 0.004 ^a^
	Chromium (Cr)	0.095 ± 0.005 ^a^	0.180 ± 0.079 ^a^
	Cadmium (Cd)	0.007 ± 0.003 ^a^	0.006 ± 0.002 ^a^
	Lead (Pb)	0.039 ± 0.019 ^b^	0.123 ± 0.008 ^a^
Nitrate	NO_3_^−^	778.88 ± 54.95 ^a^	701.2 ± 33.72 ^a^

Note: Different letters indicate significant differences in content between wild and cultivated *Z. striolatum* (*p* < 0.05); and “DW” means dry weight.

**Table 2 plants-12-02180-t002:** The content of free amino acids of wild and cultivated *Zingiber striolatum* (mg/g DW).

Compounds	Wild	Cultivated
Aspartic acid (Asp)	1.070 ± 0.052 ^a^	0.999 ± 0.096 ^a^
Serine (Ser)	2.199 ± 0.055 ^a^	1.697 ± 0.051 ^b^
Glutamic acid (Glu)	0.804 ± 0.094 ^a^	1.082 ± 0.110 ^a^
Glycine (Gly)	0.244 ± 0.006 ^b^	0.319 ± 0.004 ^a^
Histidine (His)	3.974 ± 0.093 ^a^	1.557 ± 0.034 ^b^
Arginine (Arg)	0.324 ± 0.040 ^b^	0.426 ± 0.007 ^a^
Threonine (Thr) *	0.273 ± 0.021 ^a^	0.177 ± 0.012 ^b^
Alanine (Ala)	0.293 ± 0.012 ^a^	0.276 ± 0.005 ^a^
Proline (Pro)	1.020 ± 0.091 ^a^	0.477 ± 0.063 ^b^
Cysteine (Cys)	0.011 ± 0.003 ^a^	0.013 ± 0.006 ^a^
Tyrosine (Tyr)	0.162 ± 0.010 ^a^	0.058 ± 0.030 ^b^
Valine (Val) *	0.365 ± 0.006 ^a^	0.126 ± 0.016 ^b^
Methionine (Met) *	0.053 ± 0.004 ^a^	0.024 ± 0.005 ^b^
Lysine (Lys) *	0.127 ± 0.006 ^a^	0.088 ± 0.004 ^b^
Isoleucine (Ile) *	0.238 ± 0.010 ^a^	0.057 ± 0.001 ^b^
Leucine (Leu) *	0.265 ± 0.003 ^a^	0.091 ± 0.010 ^b^
Phenylalanine (Phe) *	0.170 ± 0.008 ^a^	0.053 ± 0.036 ^b^
EAA (%)	12.85	8.19
Total	11.592 ± 0.383	7.521 ± 0.172

Note: Different letters indicate significant differences in content (*p* < 0.05); “*” means essential amino acids; and “DW” means dry weight.

**Table 3 plants-12-02180-t003:** Antioxidant activity of wild and cultivated *Zingiber striolatum*.

Activity	Wild	Cultivated
DPPH (mg/g DW)	16.47 ± 0.10 ^a^	17.20 ± 0.15 ^a^
ABTS^+^ (mg/g DW)	19.98 ± 0.26 ^a^	21.97 ± 0.10 ^a^
FRAP (μmol/L)	59.69 ± 1.45 ^b^	72.07 ± 4.78 ^a^

Note: Different letters indicate significant differences in content (*p* < 0.05); “DW” means dry weight.

**Table 4 plants-12-02180-t004:** Hypoglycemic activity of wild and cultivated *Zingiber striolatum*.

Hypoglycemic Activity		Inhibitory Activity (%)			
		Wild	Cultivated	60% Ethanol	Acarbose
α-Glucosidase	Alcohol extraction	98.72 ± 0.81 ^a^	98.66 ± 1.72 ^a^	46.10	57.80
	Water extraction	-	-	/	98.98
α-Amylase	Alcohol extraction	78.28 ± 4.02 ^a^	81.52 ± 4.65 ^a^	52.71	87.99
	Water extraction	53.26 ± 3.82 ^a^	43.97 ± 6.73 ^a^	/	83.67

Note: Different letters indicate significant differences in content (*p* < 0.05) between wild and cultivated *Zingiber striolatum*; “-” means no significant inhibitory activity detected; and “/” means no mock with 60% ethanol need to do.

**Table 5 plants-12-02180-t005:** Volatile composition of wild and cultivated *Zingiber striolatum*.

No	Retention Time (min)	Molecular Formula	Compound	Relative Content (%)	
				Wild	Cultivated
1	5.541	C_10_H_16_	α-Pinene	0.96	1.65
2	6.233	C_10_H_16_	β-Pinene	2.67	5.30
3	6.661	C_10_H_16_	α-Phellandrene	-	0.17
4	7.069	C_10_H_16_	Thujene	2.54	-
5	7.071	C_10_H_16_	β-Phellandrene	-	5.23
6	9.964	C_9_H_14_O	4-isopropylcyclohex-2-en-1-one	0.08	0.15
7	14.278	C_15_H_24_	β-Elemene	1.03	2.21
8	14.957	C_15_H_24_	Caryophyllene	0.41	0.95
9	15.182	C_15_H_24_	γ-Elemene	-	0.22
10	15.741	C_15_H_24_	Humulene	1.11	2.68
11	16.497	C_15_H_24_	β-Selinene	-	0.10
12	18.685	C_15_H_24_O	Caryophyllene oxide	-	0.29
13	19.283	C_15_H_24_O	(-)-Humulene epoxide II	-	0.46
14	22.792	C_16_H_32_O_2_	Ethyl myristate	0.20	-
15	24.741	C_17_H_34_O_2_	Methyl 14-Methylpentadecanoate	0.31	-
16	24.559	C_15_H_26_O	(-)-Isolongifolol	-	0.30
17	24.739	C_17_H_34_O_2_	Methyl palmitate	-	0.33
18	25.373	C_16_H_32_O_2_	Palmitic acid	4.07	2.65
19	25.59	C_18_H_36_O_2_	Palmitic acid ethyl ester	15.00	6.31
20	26.756	C_21_H_44_	Heneicosane	0.68	0.63
21	26.817	C_19_H_32_O_2_	Methyl linolenate	0.34	0.45
22	27.361	C_20_H_36_O_2_	Linoleic acid	-	3.09
23	27.451	C_20_H_36_O_2_	Linoleic acid ethyl ester	14.63	-
24	27.532	C_20_H_34_O_2_	Linolenic acid ethyl ester	21.20	16.11
25	27.629	C_20_H_36_O_2_	Ethyl linoleate	3.72	1.40
26	27.745	C_20_H_40_O_2_	Ethyl stearate	3.23	-
27	27.944	C_18_H_32_O_2_	Linoleicacid	1.02	-
28	28.559	C_23_H_46_	cis-9-Tricosene	0.23	0.15
29	28.753	C_23_H_48_	Tricosane	5.16	4.66
30	29.081	C_19_H_36_	5-Butyl-6-hexyloctahydro-1H-indene	0.15	-
31	30.46	C_25_H_52_	Pentacosane	0.30	-
32	30.86	C_20_H_34_O	Thunbergol	3.77	3.56
33	29.608	C_22_H_38_	1,1′-Ethylenebisdecalin	-	19.22

Note: “-” means not detected.

## Data Availability

The data presented in this study are available.

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
