# Peer review of "Wild vs. Cultivated Zingiber striolatum Diels: Nutritional and Biological Activity Differences"

_plants, 2023, doi:10.3390/plants12112180_

Round 1

Reviewer 1 Report

1.  INTRODUCTION

Lines 41-42: Zingiber striolatum belongs to Zingiber species, and is a unique medicinal and food plant resource in China[4]

- only medicinal and food source in China? Is the sentence true?  Possibly, th3 reference [4], line 590,  does not justify this sentence!

2  RESULTS  - Line 64

2.1. Chemical Composition

2.1.1 Moisture, soluble protein and soluble sugars

Lines 67 to 76: Figure 1 showed the moisture (a), soluble protein (b) and soluble sugar (c) values

The determination of the chemical composition referring to moisture, soluble protein and soluble sugars was done judiciously, however, very detailed, causing doubts from a practical point of view.

5. Conclusions - p. 546

Figure 8, p. 568 is not outlined in a comprehensible way

Author Response

Response to Reviewer 1 Comments

 Point 1: Lines 41-42: Zingiber striolatum belongs to Zingiber species, and is a unique medicinal and food plant resource in China[4]- only medicinal and food source in China? Is the sentence true?  Possibly, th3 reference [4], line 590,  does not justify this sentence!

Response 1: Thank you for your comments. We have changed “unique” to “special” and hopefully it is a more accurate description.

 Point 2: 2.1. Chemical Composition

2.1.1 Moisture, soluble protein and soluble sugars

Lines 67 to 76: Figure 1 showed the moisture (a), soluble protein (b) and soluble sugar (c) values

The determination of the chemical composition referring to moisture, soluble protein and soluble sugars was done judiciously, however, very detailed, causing doubts from a practical point of view.

Response 2: Thank you for your comments. We made some appropriate deletions to make the text more concise.

 Point 3:  Figure 8, p. 568 is not outlined in a comprehensible way

Response 3: Thank you for your comments. We have made some optimizations to the images and descriptions.

Reviewer 2 Report

Commentes

First, in title and plant material the specie must be named correctly:  Zingiber striolatum Diels

In paragraph 103-107:  “The results in this work showed that Cr and Pb levels in wild materials were below the corresponding thresholds for the highest levels of food contaminants (fresh vegetable, Cr≤0.5 mg/kg FW and Pb≤0.1 mg/kg FW) in China (GB 2762-2017), however, the Cr and Pb content of cultivated Z. striolatum were 1.15-fold and 3.92-fold higher than the standard, respectively. The reason for the excess contaminants may be due to the greater ability of the species to retain this metal”

Comments: When the authors give a reason for the excess of contaminants, it is speculative, since they have not previously described the conditions in which the cultivated material was developed, please justify or in the conclusion highlight this aspect of the toxic content, as a recommendation to cultivate the species.

In paragraph 147-148:  The amino acid results for Z. striolatum in this study are not likely to be comparable to those in the literature, as there are fewer studies available.

Comments: Must be included the article/les or cite authors

Run 216   2.2.2. Hypoglycemic activity

The authors evaluated α-glucosidase and α-amylase inhibitors in vitro, the hypoglycemic activity is a potential data. I suggest change this as:

2.2.2. α-glucosidase and α-amylase inhibition

Figure 6 and 7. I think that no is necessary, suggest delete, because the same data in table 6. or can be included as supplementary material

In order to improve, I suggest including in discussion a paragraph that relates volatiles and bioactivities

4. Materials and Methods

 (run 375-377), 4.2.2. Soluble protein.. “…and then the content of soluble protein 375 was calculated according to the standard curve..”

Please, included standard used, and revise to each assay

Author Response

Response to Reviewer 2 Comments

Point 1: First, in title and plant material the specie must be named correctly:  Zingiber striolatum Diels

Response 1: Thank you for your comments. We have corrected it in title and plant material. 

Point 2: In paragraph 103-107:  “The results in this work showed that Cr and Pb levels in wild materials were below the corresponding thresholds for the highest levels of food contaminants (fresh vegetable, Cr≤0.5 mg/kg FW and Pb≤0.1 mg/kg FW) in China (GB 2762-2017), however, the Cr and Pb content of cultivated Z. striolatum were 1.15-fold and 3.92-fold higher than the standard, respectively. The reason for the excess contaminants may be due to the greater ability of the species to retain this metal”

Comments: When the authors give a reason for the excess of contaminants, it is speculative, since they have not previously described the conditions in which the cultivated material was developed, please justify or in the conclusion highlight this aspect of the toxic content, as a recommendation to cultivate the species.

Response 2: Thank you for your comments. We have added suggestions for cultivating this species in our conclusion. 

Point 3: In paragraph 147-148:  The amino acid results for Z. striolatum in this study are not likely to be comparable to those in the literature, as there are fewer studies available.

Comments: Must be included the article/les or cite authors

Response 3: Thank you for your comments. We have made some changes to this section.

Point 4: Run 216   2.2.2. Hypoglycemic activity

The authors evaluated α-glucosidase and α-amylase inhibitors in vitro, the hypoglycemic activity is a potential data. I suggest change this as:

2.2.2. α-glucosidase and α-amylase inhibition

Response 4: Thank you for your comments. We have made corrections to the subtitle.

Point 5: Figure 6 and 7. I think that no is necessary, suggest delete, because the same data in table 6. or can be included as supplementary material

In order to improve, I suggest including in discussion a paragraph that relates volatiles and bioactivities

Response 5: Thank you for your comments. We have deteted Figure 6 and 7 in the text and  upload as supplementary materials. Furthermore, some discussions on volatiles and biological activities were added in section “Discussion”.

Point 6: (run 375-377), 4.2.2. Soluble protein.. “…and then the content of soluble protein 375 was calculated according to the standard curve..”

Please, included standard used, and revise to each assay

Response 6: Thank you for your comments. We have described the practice of standard curves.

Reviewer 3 Report

The manuscript: Wild vs cultivated Zingiber striolatum: nutritional and functional differences, describes some details about the differences among the wild and cultivated samples of Z. striolatum regarding its chemical content.

The paper is well described in several aspects, but the authors should be consider some details to improve the quality in its content, because there are some aspects they did not consider:

i. some references are missing or do not correspond to the topic

ii. In the analysis of total phenolic compounds the values are smallest than those of total flavonoids, and the authors did not explain the reason why this pattern.

iii. In the title of the paper are involved functional differences, but in the text it is not justified such relation.

There are some words written wrong

Author Response

Response to Reviewer 3 Comments

Point 1: some references are missing or do not correspond to the topic

Response 1: Thank you for your comments. We have rechecked the references.

 Point 2: In the analysis of total phenolic compounds the values are smallest than those of total flavonoids, and the authors did not explain the reason why this pattern.

Response 2: Thank you for your comments. We have explained in the discussion as follow: “Usually, flavonoids are a type of phenolic compound, and their contents are generally lower than that of total phenols, but different conclusions have been drawn in our study. We speculate that this phenomenon may be caused by different extractants. Mohammad et al. [36] studied the total phenols and flavonoids in Brassica vegetables using different solvents. The results showed that the total phenolic content measured by the methanol extract of cauliflower was much lower than the total flavonoid content measured by the ethanol extract. Their results are similar to our study and may better explain this phenomenon.”

 Point 3: In the title of the paper are involved functional differences, but in the text it is not justified such relation.

Response 3: Thank you for your comments. We have made changes to the title. It was changed to ''Wild vs cultivated Zingiber striolatum Diels: nutritional and biological activity differences''.

Point 4: There are some words written wrong

Response 4: Thank you for your comments. We have checked the spelling of the words carefully.

Round 2

Reviewer 2 Report

Point 1: First, in title and plant material the specie must be named correctly:  Zingiber striolatum Diels

Response 1: Thank you for your comments. We have corrected it in title and plant material.

 Reponse REW point 1:  Only in title and plant material the specie must be named correctly:  Zingiber striolatum Diels,  While in the rest of document, can be namer Z. striolatum, Please revise

Author Response

Response to Reviewer 2 Comments

Point 1:  Reponse REW point 1:  Only in title and plant material the specie must be named correctly:  Zingiber striolatum Diels,  While in the rest of document, can be namer Z. striolatum, Please revise

Response 1: Thank you for your comments. It is revised.
